# The Emerging Role of CD24 in Cancer Theranostics—A Novel Target for Fluorescence Image-Guided Surgery in Ovarian Cancer and Beyond

**DOI:** 10.3390/jpm10040255

**Published:** 2020-11-27

**Authors:** Katrin Kleinmanns, Vibeke Fosse, Line Bjørge, Emmet McCormack

**Affiliations:** 1Center for Cancer Biomarkers CCBIO, Department of Clinical Science, University of Bergen, Jonas Lies vei 91B, 5021 Bergen, Norway; Vibeke.Fosse@uib.no (V.F.); Line.Bjorge@uib.no (L.B.); Emmet.Mc.Cormack@uib.no (E.M.); 2Department of Obstetrics and Gyneacology, Haukeland University Hospital, 5021 Bergen, Norway; 3Centre for Pharmacy, Department of Clinical Science, University of Bergen, Jonas Lies vei 65, 5021 Bergen, Norway; 4Department of Clinical Science, University of Bergen, Jonas Lies vei 65, 5021 Bergen, Norway

**Keywords:** biomarker, intraoperative imaging, debulking surgery, complete resection, epithelial ovarian cancer, optical imaging, CD24

## Abstract

Complete cytoreductive surgery is the cornerstone of the treatment of epithelial ovarian cancer (EOC). The application of fluorescence image-guided surgery (FIGS) allows for the increased intraoperative visualization and delineation of malignant lesions by using fluorescently labeled targeting biomarkers, thereby improving intraoperative guidance. CD24, a small glycophosphatidylinositol-anchored cell surface receptor, is overexpressed in approximately 70% of solid cancers, and has been proposed as a prognostic and therapeutic tumor-specific biomarker for EOC. Recently, preclinical studies have demonstrated the benefit of CD24-targeted contrast agents for non-invasive fluorescence imaging, as well as improved tumor resection by employing CD24-targeted FIGS in orthotopic patient-derived xenograft models of EOC. The successful detection of miniscule metastases denotes CD24 as a promising biomarker for the application of fluorescence-guided surgery in EOC patients. The aim of this review is to present the clinical and preclinically evaluated biomarkers for ovarian cancer FIGS, highlight the strengths of CD24, and propose a future bimodal approach combining CD24-targeted fluorescence imaging with radionuclide detection and targeted therapy.

## 1. Introduction

The aggressive surgical approach used to treat metastatic ovarian cancer is unique and remains one of the therapeutic keystones, with the aim of achieving curative treatment. The overall survival rate for epithelial ovarian cancer (EOC) is below 50%, mainly because the lack of early symptoms leads to a late diagnosis in almost 75% of patients, followed by limited treatment options and emerging drug resistance [1]. In 1975, Thomas Griffith was the first to show that besides the histological subtype, the extent of residual disease after surgery impacts patient outcome [2]. The extent of residual disease is classified into complete cytoreduction (0 cm) and macroscopic residual disease, which is further subclassified into optimal (1 cm) and suboptimal (1 cm) cytoreduction [3]. The impact of complete debulking has been confirmed in many trials as the most important prognostic factor for the survival of EOC patients, and it has been suggested to increase the efficacy of subsequent drug therapies [3,4,5]. Clinical trials have also indicated that if macroscopic residual disease remains after surgery, the patient has no significant benefit compared to those who received suboptimal debulking, highlighting the need for better tools to help surgeons achieve complete resection.

In clinical reality, complete debulking is not always feasible and can be challenging because of aggressive tumor biology, histological subtype, advanced disease stage, and unresectable tumors near vital structures [6,7,8]. To detect submillimeter and residual tumor lesions, the surgeon relies on tactile and visual inspection, training, and experience. New technologies for preoperative planning are warranted to help identify patients who will benefit from surgery, and for these patients, intraoperative guidance methods to help achieve complete debulking are crucial for their survival.

One surgical technology that was developed to meet these criteria is fluorescence image-guided surgery (FIGS). FIGS employs fluorescent molecules (fluorophores) as contrast agents, and together with specialized imaging systems, the surgeon is provided with a screen image highlighting fluorescent tumor tissue in the operating field. This technology has the capability to provide the surgeon with real-time feedback and optimize the precision of resection, thus improving clinical outcomes. FIGS is currently being assessed in ovarian cancer clinical trials, as well as for many other solid malignancies. Ovarian cancer Phase I–III FIGS trials are being performed with untargeted dyes, such as indocyanine green (ICG) [9] and contrast agents that target the folate receptor alpha (FRα). The first in-human FIGS study was performed by Go van Dam et al. in 2011, in which they demonstrated the feasibility of FIGS to improve the visualization of malignant lesions with a tumor-specific FRα-targeting probe, which had high specificity for peritoneal metastases and left cancer-free tissue fluorescence negative [10]. In fact, this tumor-specific fluorescence approach resulted in an increased resection rate for malignant tissue. Since this first pilot study, FRα-targeting FIGS contrast agents have been improved, and there are promising results from Phase II [11,12] and ongoing Phase III (NCT03180307) clinical trials. To expand beyond folate and other clinically approved monoclonal antibodies (mAbs), much effort is being dedicated to identifying new biomarkers for FIGS. Notably, the challenge faced when developing a good imaging contrast agent for FIGS is the identification of both a biomarker that can discriminate between malignant and healthy tissues and a dye capable of maximal tissue penetration which can be detected with a dedicated and sensitive fluorescence imaging system. The biomarker-dye conjugate also needs to exhibit low toxicity, high stability, and fast clearance to reduce unspecific signals [13].

Cluster of differentiation 24 (CD24) is a novel molecular target used for imaging, molecular-targeted drug therapy, and immunotherapy [14,15,16]. Its high expression on tumor cells and rare expression on healthy human tissue, where it is primarily found on hematologic cells, indicate its potential as a tumor-specific biomarker [14,15,16]. A meta-analysis of 28 studies revealed that CD24 is overexpressed in 68% of human cancers, and CD24 expression was correlated with a higher self-renewal ability, more metastases, and a poor prognosis [17]. In EOC, CD24 is almost uniformly (70.1–100%) expressed [18,19]. We have demonstrated the ability of a CD24-targeting mAb conjugated with near-infrared (NIR) fluorophores to identify tumor lesions in metastatic EOC patient-derived xenograft (PDX) models with heterogeneous CD24 expression, supporting its potential to translate CD24-guidance to the intraoperative setting for EOC patients [14,19]. Here, we review the emerging tumor-specific biomarker CD24 as a target for FIGS, and focus specifically on EOC and the use of CD24 as a bimodal biomarker for both imaging and theranostics.

## 2. Fluorescence Image-Guided Surgery

FIGS is a new strategy with the potential to (1) enhance the visualization of submillimeter metastases, (2) improve tumor staging and risk stratification, (3) increase contrast between malignant and healthy tissues, (4) optimize negative resection margins, (5) minimize damage to healthy structures, and (6) confirm tumor cell-free resection beds intraoperatively in (7) real-time [20]. In a successful debulking surgery with complete removal, all malignant lesions are resected, tumor cell-infiltrated lymph nodes are removed, and negative tumor margins are achieved without damaging vital and healthy structures [20]. In EOC clinical trials focused on the impact of residual disease, complete debulking surgery (0 mm residual disease) was achieved in 33.5–37% of cases, while optimal debulking surgery was achieved in 64.7–74% of cases [3,4,5]. In addition, among several different solid cancers, positive tumor margins occur in 8–70% of tumor resections, which correlates with local recurrence and a poor prognosis [13,21,22]. Although FIGS appears to be an ideal emerging technology to help prevent incomplete tumor surgeries, and several imaging probes have been shown to allow differentiation between malignant and healthy tissues, the probes often fail to achieve sufficient contrast to clearly delineate the tumor margins. This highlights the challenges facing the development of sensitive and specific FIGS probes—the identification of compatible and suitable dyes, biomarkers, and imaging technologies that together allow the specific targeting and intraoperative identification of malignant lesions [23].

Dyes currently being investigated in fluorescence optical imaging technologies for clinical use have shifted from those that emit in the visible light spectrum to dyes that emit in the NIR spectrum (700–900 nm) [13,20]. The NIR spectrum offers reduced tissue autofluorescence, scattering, and tissue absorption characteristics, therefore allowing deeper tissue penetration with better contrast compared to the visible light spectrum (400–700 nm). Recent advances in NIR-II (900–1700 nm) fluorescence probes (e.g., nanotubes) permit even deeper tissue penetration, a larger Stokes shift, and less photobleaching than with organic dyes, achieving a superior resolution in the micrometer range [24]. Fluorescent contrast agents offer the advantages of nonionizing properties and high temporal resolution. To date, the only FDA- and EMA-approved NIR fluorescent contrast agents for FIGS are ICG (λ_ex_ 783, λ_em_ 813) and methylene blue (MB, λ_ex_ 670, λ_em_ 690). These contrast agents can accumulate passively in tumors because of the enhanced permeability and retention (EPR) effect caused by leaky blood vessels and high interstitial pressure, but they are not targeted; therefore, they are not specific enough to reliably guide tumor resection in most cancers [25].

Coupling NIR fluorophores to targeting molecules increases tumor specificity and decreases false-positive signals. The targeting moiety can be an antibody, antibody fragment, peptide, or small molecule; they vary in size from large macromolecules to small nanoparticles and accumulate in tumor tissue through specific binding to antigens present on tumor cells or cells in the tumor microenvironment [26]. The molecular size of the binding moiety influences the half-life circulation of the targeting molecule and therefore determines the time point of surgery, with optimal fluorescence signal intensity obtained from between 2 and 8 hours for small molecules and peptides, to 48–72 h for antibodies [26]. To achieve this accumulation, the targeting molecule must first pass certain barriers in the body to penetrate the tumor and reach its specific biomarker. The pharmacokinetic properties of the probe are strongly influenced by the fluorophore, charge, labeling method, and solubility [27]. Next, to release a specific signal at the tumor, the fluorescent contrast agent has to bind the abundantly expressed receptor with high affinity and not undergo unspecific accumulation elsewhere in the body. Third, unbound molecules have to be rapidly cleared from the circulation through the liver and kidneys to enhance contrast. Therefore, tumor biomarkers eligible for FIGS must meet the criteria of highly specific tumor uptake, low unspecific binding, and rapid clearance. Together with the dye, the conjugate has to exhibit minimal toxicity and high plasma stability. Notably, tumor lesion sensitivity, which is necessary to produce the optimal signal-to-background ratio with the highest contrast, is dependent on both the fluorescent and binding moieties [20].

Ultimately, emitted fluorophores have to be detected by fluorescence-guided surgery systems with nanomolar-level sensitivity [28]. In open surgery systems, the detected fluorescent signal is displayed on a screen, which means that FIGS does not interfere with the standard surgical field or operative procedures. Other types of imaging systems also exist, such as fluorescence detection integrated into endoscopic devices and robotic surgery systems [20]. To preserve the photostability of the fluorophores, these camera systems are designed to keep the fluorescence excitation rate below the photobleaching threshold of NIR dyes, avoiding irreversible photobleaching and allowing a long-lasting intraoperative fluorescence signal [29]. In addition, to facilitate the translation of the fluorescence signal displayed on a screen to the operation field, wide-field-of-view imaging goggles (goggle navigation system) with an integrated overlay function represent a possible future solution [30]. Multiple factors, such as tissue optical properties, depth, diffused fluorescence resolution, illumination uniformity, and quantification of the fluorescence signal, influence imaging system performance and acquired fluorescence intensities, making it difficult to standardize imaging systems [31]. Therefore, quality control procedures for imaging performance and the quantification of fluorescence data are necessary [32]. Emerging computational calculation methods can be exploited to process the large amount of data and correct for autofluorescence and scattering. A technology based on fluorescence lifetime analysis is capable of differentiating between true-positive and false-positive signals on the basis of distinct fluorescent decay after excitation of the fluorophores [33]. This decay is independent of fluorophore concentration, tissue scattering, and tissue depth.

In recent years, the number of early phase trials using FIGS has markedly increased. Because of advances in (1) fluorescence imaging systems that detect, process, and quantify fluorescence, (2) fluorescent contrast agent design (e.g., activatable probes, nanomolecular probes), and (3) the use of fluorescent imaging beyond real-time guidance in combination with photothermal therapies, immunotherapy, and antibody–drug conjugates, FIGS has the potential to revolutionize the treatment and improve the outcomes of cancer patients.

## 3. Biomarkers Employed in EOC

To date, most biomarkers tested in clinical trials of FIGS are clinically approved mAbs evaluated for molecular targeted therapy [13]. These GMP-manufactured antibodies have been found to be safe, are readily available, and are more likely to become approved for clinical use. A biomarker (biological marker) is “a characteristic that is objectively measured and evaluated as an indicator of normal biological processes, pathogenic processes, or pharmacologic responses to a therapeutic intervention” [34]. Clinical biomarkers can be categorized as diagnostic, prognostic, or predictive. Biomarkers that are tumor-specific and sensitive are needed to design targeted therapies; of particular value are monoclonal antibodies because they selectively target tumor cells with high affinity and are thus less cytotoxic to healthy tissue, unlike traditional chemotherapy in most cases. MAbs that target biomarkers of EOC, such as cancer antigen 125 (CA125), epithelial cell adhesion protein (EpCAM), vascular endothelial growth factor (VEGF), epidermal growth factor receptor (EGFR), and folate receptor alpha (FRα), have been investigated in clinical trials of a molecular targeted therapy alone or in combination with standard of care and have been extended following fluorophore conjugation to FIGS (Table 1).

In EOC, an increase in the serum level of CA125 is often observed before clinical symptoms occur, representing the current gold standard for diagnosing and monitoring the treatment response and disease progression. However, serum CA125 is not tumor-specific, varies between patients, and can increase in response to various physiological conditions without neoplastic involvement, such as endometriosis [50]. Targeting the CA125 surface antigen, which is expressed in more than 95% of all non-mucinous EOC, has been suggested as a target for immunotherapy [35]. Despite CA125 being almost ubiquitously expressed in epithelial ovarian tumors, and the most frequently used diagnostic biomarker of EOC, this approach for immunotherapy has not shown a clinical benefit [35]. For FIGS, however, CA125 is a potential tumor-specific biomarker [36]. The preclinical study by Fung et al. explored the CA125-targeting antibody B43.13 conjugated to IR800 in orthotopic OVCAR3 and PDX models, and found low background signals in healthy organs and specific uptake in CA-125-positive PDX and primary tumors [36].

The accumulation of ascites due to increased vascular permeability and the early onset of tumor angiogenesis is mediated by vascular endothelial growth factor (VEGF) overexpression [51]. The clinically approved (in 2011) mAb bevacizumab targets the tumor-promoting and immunosuppressive molecule VEGF-A. Inhibition of the VEGF receptor results in decreased accumulation of ascites, and increased progression-free survival and patient quality of life [51]. The family of epidermal growth factor receptor (EGFR)-targeted therapies, including trastuzumab, pertuzumab and cetuximab, has been investigated in EOC patients, finding only moderate effects. Trastuzumab targets the extracellular domain of HER2, which blocks tumor-associated signaling pathways believed to reduce tumorigenicity and invasiveness and activates antibody-dependent cellular cytotoxicity (ADCC). The low rate of HER2 overexpression, in only 11.4% of cases, together with the low response rate reported in the first Phase II led to the conclusion that HER2 is not a suitable target for EOC [38]. A preclinical evaluation of the targeting of VEGF and HER2 in intraperitoneal ovarian cancer cell line models with IRDye800CW-labeled bevacizumab and trastuzumab reported specific tumor detection in vivo, suggesting its potential as a targeting moiety for image-guided surgery [37]. Of note, the dose-dependent tumor-to-background ratios were 1.93 (bevacizumab) and 2.92 (trastuzumab), and it was possible to detect submillimeter metastases, although there were high levels of unspecific signaling [41]. Nanobodies offer improved tissue penetration and rapid blood clearance, and are therefore an attractive technology for the development of targeted imaging probes. A HER2-targeting NIR-labeled nanobody has been developed, and preclinically, it was demonstrated to enable the fast and highly sensitive detection of submillimeter-sized lesions with decreased false-positive detection [42]. However, these experiments were performed on HER2^+^ Skov-3 intraperitoneally injected cell line models, and given the wide variation in HER2 expression in ovarian cancer, more homogenously expressed biomarkers are required to further develop this approach.

The trifunctional mAb catumaxomab targets CD3 on T-cells, Fc-receptors on innate immune cells, and EpCAM on tumor cells, and it is heterogeneously overexpressed in epithelial cancers, including EOC. It has been tested for molecular target therapy in Phase I–III EOC clinical trials. The results of these studies indicate its efficacy in the management of ascites, but with only partial tumor response in 5% of patients [44,45]. Disregarding its high abundance on epithelial tumor cells, EpCAM is also expressed on healthy human epithelia, which is a suboptimal characteristic for an imaging biomarker. Yet, EpCAM-IRDye800CW has been investigated in preclinical models of colon, breast, and head and neck cancers, and it was found to aid in the identification of millimeter-sized metastases that were missed by the naked eye [52].

The most thoroughly investigated biomarker for use in FIGS with EOC is FRα. FRα, also known as FOLR1, has been reported to be expressed on 76% of histological EOC samples. The expression level is dependent on the histologic subtype (highest in high-grade serous ovarian cancer (HGSOC)), FIGO stage (increases with stage), and recurrence status (lower in primary EOC) [53,54]. Its low abundance in nonmalignant tissues makes FRα an attractive target for several applications in EOC—molecular targeted therapy with the mAb farletuzumab and the drug-conjugate vintafolide [54], targeted photothermal therapy with IR780 NIR light irradiation [49], and FRα-targeted real-time fluorescence imaging in debulking surgeries [10]. Molecular targeted therapy with farletuzumab was shown to have a promising objective response rate of over 70% in a Phase II trial, but a randomized Phase III trial with 1100 participants failed to show a benefit on median progression-free survival [46,55]. In contrast, after the introduction of the first FRα-targeted fluorescence imaging probe to the intraoperative setting, which enabled the tumor-specific visualization of ovarian cancer metastases during surgery, folate-FITC (EC17) was evaluated in a larger EOC patient cohort [10,11]. Intraoperative guidance with EC17 allowed the additional detection of 16% more metastases not identified by visual inspection and palpation alone [11]. The fluorescence signal was specific in FRα-positive lesions, but high false-positive detection rates occurred due to tissue autofluorescence, as the fluorophore is in the visible fluorescence spectrum. To overcome this problem, folate was conjugated to a NIR fluorophore (OTL38), and in Phase II EOC clinical trials, resection rates improved by 29% additionally detected fluorescent lesions [48], with 48.3% of patients deriving a benefit from the superior detection of at least one extra metastasis [12]. In both studies, false-positive signals (23% [32] and 10% [12]) were attributed to specific binding to folate receptor β, which is expressed, for instance, on activated macrophages in lymph nodes, and demonstrates that there is still a need for a more specific imaging probe for EOC patients.

The implementation of clinically approved biomarkers in ovarian cancer as a target for FIGS has demonstrated the feasibility and benefit of intraoperative fluorescence guidance for cytoreductive surgeries. Apart from clinically evaluated ovarian cancer biomarkers, promising preclinical studies are examining new biomarkers, such as integrins [56] and the gonadotropin-releasing hormone (GnRH) receptor [57], which are not only able to detect minuscule tumor deposits but are also uniformly expressed in many cancers, and thus widen the applicability of these imaging contrast agents. The overexpression of the (α_v_β_3_) integrin in many cancers, its low expression in normal tissues (except kidney), and its internalization after binding the FIGS contrast agent were shown to result in slower clearance and specific tumor uptake in subcutaneous ovarian A2780 and OVCAR-4 cell line models [56,58]. These studies suggest that integrins, targeted by IntegriSense 680 and cRGD-deep-red squaraine fluorophore, are promising targets for FIGS in EOC and other cancers, and breast cancer integrin-targeted FIGS has already been used in clinical trials [13]. The GnRH antagonist peptide conjugated to ICG has also been evaluated for preclinical FIGS [57]. The GnRH receptor is expressed on 78% of EOC and on other hormone-related cancers, such as endometrial, breast and prostate cancers. A study by Liu et al. reported high-affinity binding to GnRH receptor-expressing EOC cells and a high tumor-to-background ratio, but also an increased fluorescence signal intensity in healthy organs compared to ICG alone in metastatic xenograft models [57].

## 4. CD24 Is Classified as a Cancer-Specific and Sensitive Biomarker for Fluorescence Image-Guided Surgery

One of the preclinically evaluated and promising biomarker candidates is CD24. CD24 is a small, heavily glycosylated mucin-like glycosylphosphatidylinositol (GPI)-linked cell surface protein that is expressed on developing and regenerating tissues. CD24 plays a crucial role in the early maturation during hematopoiesis and is physiologically expressed on premature lymphocytes (pre-B-cells), granulocytes (neutrophils), some epithelial (e.g., keratinocytes, renal tubular epithelial cells), neural, and pancreatic cells [59]. In contrast to its low expression in healthy tissues, the adhesion molecule CD24 is almost ubiquitously expressed in EOC (70.1–100%) and in 68% of solid tumors, for many of which surgical resection is a main therapeutic strategy, such as colorectal cancer, advanced pancreatic cancer and gliomas [17] (Table 2). Its primarily membranous expression is advantageous, as it permits superior antigen targeting without the need to be internalized into the cells. CD24 was found to be membranously expressed in approximately 90% of EOC patients and in 100% of established orthotopic EOC PDX models [19]. The low expression in healthy tissues together with its overexpression in various malignant tumors (Table 2), suggest its tumor specificity and broad clinical applicability. As a result, CD24 overexpression has been exploited as a diagnostic marker and as a marker for treatment resistance and prognosis, and evaluated for targeted therapy [17,60,61].

In recent years, CD24 has evolved as an encouraging molecular biomarker for EOC [78] in the field of immunotherapy, and as a tumor-specific biomarker for targeted drug delivery and imaging [14,16,19]. CD24 expression in ovarian cancer has been found to correlate with clinicopathological parameters, such as higher tumor grade, higher tumor stage, omental metastasis, relapse, and a more aggressive course of the disease [17,66,79]. As already mentioned, complete debulking surgery is one of the most important prognostic factors in EOC, with emerging evidence that patients with no macroscopic residual tumor deposits have an increased survival benefit [2,3,4,5]. Based on available data bases, complete debulking has a similar survival benefit in early-stage and advanced-stage disease [3]. The detection of metastases is additionally important during cytoreductive surgery to improve the staging of the patient for adequate postsurgical treatment and to avoid radical surgery that may lead to complications. In addition, a significant survival benefit with secondary cytoreductive surgery in recurrent EOC was reported in the DESKTOP III trial in patients who underwent complete debulking surgery [80] (and presented at ASCO2020). Studies have shown that fluorescence-guided surgical procedures can lead to changes in the postsurgical treatment plan, mainly because of the identification of additionally identified peritoneal and unresectable metastases [81].

CD24-targeted FIGS can improve the identification of disseminated metastases based on the association of CD24 expression with an advanced stage and the metastatic activity of CD24^+^ tumor cells, which exhibit fast tumor growth [17,82]. The increased expression of CD24 in metastatic tumors was identified in 60 paired primary and metastatic lesions of bladder cancer [62]. We have demonstrated that CD24-Alexa Fluor 680 can positively identify tumor lesions through non-invasive imaging at an early stage of the disease, detect early metastatic dissemination, and allow the longitudinal evaluation of tumor progression and treatment efficacy [19]. For real-time intraoperative fluorescence imaging, conjugation to the NIR dye Alexa Fluor 750 improved its optical properties by allowing deeper tissue penetration along with low tissue autofluorescence. FIGS enabled the intraoperative identification of the primary tumor and miniscule metastases in seven heterogeneous CD24-expressing PDX models, and ultimately improved tumor resection in preclinical orthotopic PDX [14]. CD24’s monoclonal antibody-based fluorescence intensity peaked 48 h after injection. However, a tumor-specific fluorescence signal was obtained for up to 120 h, demonstrating the long retention time of the contrast agent [14]. CD24-targeting contrast agents can be further developed into smaller antibody fragments, such as nanobodies and minibodies, which have favorable pharmacokinetic characteristics, while maintaining their superior binding affinity [13,83]. Of note, in addition to the HGSOC subtype, we extended their use to clear cell carcinoma and anaplastic carcinoma models, demonstrating the versatility of CD24-targeted fluorescence imaging [14,19]. Our results suggest that CD24 is a promising tumor target for FIGS in ovarian cancer patients, as it fulfils the requirements of a good biomarker for the development of FIGS contrast agents (Figure 1). However, the increased expression of CD24 in metastases compared to that in its matched primary lesion has not been confirmed for EOC.

Aside from the overexpression of the target on various different malignancies, homogenous biomarker expression is needed for broad clinical applicability. Notably, most tumor biomarkers, including CD24, are heterogeneously expressed (Table 2). The heterogenous spatial and temporal inter- and intratumoral expression can pose challenges for patient selection and fluorescent signal intensities in FIGS. However, to overcome intratumoral heterogeneity, multiplexing strategies involving the use of a cocktail of different contrast agents with different antigen specificities for FIGS might avoid patient preselection and increase the contrast between healthy and tumor tissues [84,85].

CD24 has not only been correlated with clinicopathological parameters in EOC, but also associated with tumor grade and overall survival in several other cancers [17,66,69,77,86] (Table 2). To understand the relationship between CD24 and poor prognosis, CD24 expression has been linked to biological characteristics. Four CD24-driven mechanisms have been described that indicate its role in carcinogenesis, and therefore might explain its tumor-specific expression. First, CD24 is an alternative ligand of P-selectin, an adhesion receptor expressed on activated endothelial cells and platelets, promoting metastatic potential through binding CD24^+^ tumor cells [87]. A direct role in metastatic progression has been described by Overdevest et al., who identified the correlation of CD24^+^ tumor cells with metastatic spread to the lungs in bladder carcinoma [62]. Second, CD24 has been associated with epithelial–mesenchymal transition (EMT), a multistep process enabling the tumor cells to invade and disseminate [66]. It has been suggested that CD24 affects the EMT signal cascade via the PI3K/Akt/mTOR pathway and MAP kinase pathways, which imparts a high invasive capacity [66]. Third, cancer stem cells (CSCs), a continuously proliferating subset of cancer cells with stem-like and tumor cell characteristics, express, among other markers, CD24, CD44, and CD133, and they increase tumorigenic potential and drive growth and metastasis. CSCs are capable of generating the full tumor heterogeneity that resembles the parental tumors, and are the main driver of an aggressive disease course; thus, they are the target of many therapeutic approaches. CD24 is a CSC marker of colorectal, pancreatic, hepatic, and nasopharyngeal cancers, as well as EOC. All tumors are amenable to debulking surgery or wide local excision, and therefore, CD24-FIGS could aid in the complete resection of peritoneal disseminated and metastatic disease [21]. Phosphorylated signal transducer and activator of transcription 3 (STAT3), which is found primarily in CD24^+^ tumor cells, promotes the self-renewal capacity of ovarian cancer CSCs, which also express higher mRNA levels of stemness genes [82,88,89]. Consequently, targeting CD24 with a monoclonal antibody in preclinical ovarian cancer cell line models has been shown to decrease cell proliferation and tumor growth [90]. Growth inhibition by inducting apoptosis following CD24-blockade has also been demonstrated in CD24-expressing colorectal tumors in vivo [91]. Lastly, cytoplasmatic CD24 competitively inhibits the binding of the tumor suppressor ARF to the nucleolar protein NPM, resulting in the inactivation and degradation of ARF, increased MDM2 levels, and decreased p53 levels. This suggests that CD24 overexpression results in the pro-tumorigenic inactivation of the tumor suppressor gene *TP53* [92]. Another study reported a high correlation between CD24 mRNA expression levels and p53 gene mutation in hepatocellular carcinoma [71]. Interestingly, the cytoplasmatic expression of CD24 is associated with poor overall survival in ovarian cancer, whereas CD24^−^CD44^+^ cells are an indicator of drug resistance [93]. It is, however, important to note that membranous and cytoplasmatic CD24 have different biological activities, and therefore comprise two independent prognostic markers. However, even though many studies suggest a direct role of CD24 in tumor growth, invasiveness and migration, and the formation of metastases, its potential molecular mechanisms are not fully understood [94].

In summary, CD24 demonstrates favorable biomarker characteristics, and when coupled with advanced fluorescent dyes and antibody technologies that offer superior resolution and contrast, CD24-targeted FIGS has the potential to improve the surgical resection of many types of tumors.

## 5. CD24 as a Bimodal Imaging Biomarker and Beyond

The association of CD24 expression with poor clinicopathological parameters suggests a direct role of CD24 in tumorgenicity. Furthermore, the specific binding of CD24-targeting fluorescence contrast agents to CD24^+^ tumor cells has been reported in preclinical metastatic cell lines and PDX models, which support its use as a theranostic biomarker [14,19]. CD24 has been investigated for use in molecular targeted therapy, highlighting the potent anti-tumor effect of CD24 blockade and the potential to unravel the blockade of the immune system, suggesting the potential of its use beyond FIGS [16,90]. However, to date, bimodal CD24-targeting approaches have not been investigated. Therefore, we propose the extension of CD24-targeted probes to (1) bimodal imaging contrast agents and (2) dual-theranostic agents that incorporate its preoperative and/or intraoperative imaging and targeted treatment characteristics.

The combination of radioactive tracers and NIR fluorescence molecules has been suggested as an approach to overcome the limited tissue penetration depth of optical fluorescence imaging while maintaining its superior spatial resolution. Even though NIR fluorophores exhibit a favorable penetration depth of 5–7 mm in the NIR spectrum (700–900 nm) compared to 1–2 mm in the visible spectrum (400–700), the detection of deeper localized metastases and preoperative staging are hampered [95]. Nuclear gamma probes can be used for deep tissue detection below 1 cm, and the surgeon can be guided via acoustic signals generated by a gamma probe counter in close proximity to the gamma radiation-emitting targeted lesions, which can then be localized and delineated by their fluorescent characteristics [96] (Figure 2A). Farletuzumab was introduced as a hybrid nuclear-fluorescent probe for FIGS in preclinical metastatic ovarian cancer xenograft models. ^111^In-farletuzumab–IRDye800CW allows the detection of deeply located tumors by radiation, while the accurate real-time delineation of tumor metastases can be achieved by FIGS [47]. In addition, bimodal probes offer preoperative tumor localization by positron emission tomography (PET) or single-photon emission computed tomography (SPECT) for surgical planning, intraoperative guidance, and postoperative treatment. The advantage of using both modalities was also demonstrated for ^89^Zr- and IRDye800CW-conjugated pertuzumab in orthotopic Skov-3 xenograft models [41]. An optimal tumor-to-background ratio was achieved for both PET and NIR fluorescence imaging after 24–72 h [41]. The circulation time of peptides and antibodies corresponds well to the half-life of the commonly used radiotracers of several hours (^64^Cu t_1/2_ 12.7 h, PET) to a few days (^111^I t_1/2_ 2.8 d, SPECT; ^89^Zr t_1/2_ 3.3d PET). The complementary use of both radioactivity and fluorescence in one injected dual-labeled tracer overcomes their individual limitations [47,96]. Moreover, photo- (immuno or thermal) therapy involves the use of a photosensitizer (e.g., Her2-affibody-NIR830, IR780-loaded folate-targeted nanoparticles), which exerts its cytotoxic effect after NIR light exposure, helping to specifically ablate unresectable or missed tumor deposits without destroying the surrounding tissue [49,97]. The CD24-NIR probe can be further developed into a bimodal probe, expanding its use beyond intraoperative fluorescence guidance (Figure 2A).

Attempts have been made to therapeutically target CD24 using antibody–drug conjugates (ADCs; mAb CD24-nitric oxide conjugate [15]), antibody-mediated cytotoxicity [90], immunotherapy (anti-CD24-CAR-NK [98]), innate immune checkpoint inhibition [16], and bispecific antibodies [99] (Figure 2B–D). The bimodal approach of fluorescence imaging and ADCs extends FIGS to include highly selective drug delivery. The CD24-linked nitric oxide (HN-01)-targeted delivery system was shown to induce apoptosis in CD24^+^ tumor cells after receptor-mediated internalization (Figure 2B)**.** Increased levels of nitric oxide were detected in the cytoplasm of CD24-overexpressing tumor cells, resulting in tumor growth suppression in preclinical hepatic carcinoma xenograft models [15]. The use of ADCs has been found to be successful in clinical trials. Mirvetuximab soravtansine is a monoclonal antibody targeting FRα that is conjugated to the cytotoxic drug maytansinoid DM4, which directly induces mitotic arrest in FRα^+^ tumors. The antitumor activity observed in preclinical studies of EOC PDX models, together with favorable tolerability profiles in patients, led to a Phase Ib trial in relapsed EOC patients (FORWARD II) [100,101]. The site-directed delivery of the cytotoxic drug in combination with bevacizumab exhibited a similar efficacy to the standard treatment of paclitaxel combined with bevacizumab in recurrent EOC, with favorable toxicity profiles [102]. Approaches to site-specific transfer drugs to tumors have also been investigated by targeting the membrane-bound GPI-linked glycoprotein mesothelin (anetumab ravtansine) and the transmembrane mucin MUC16 (CA125; sofituzumab vedotin) in EOC [103].

In addition, CD24 plays a role in immune inhibition. Aside from its interaction with P-selectin, CD24 interacts with sialic-acid-binding Ig-like lectin 10 (Siglec-10), a receptor expressed on tumor-associated macrophages (TAMs) [16,104]. Cancer cells evade phagocytosis by TAMs through the expression of the “don’t eat me”-receptors CD47, PD-L1, MHC-I β2 microglobulin subunit (b2m), and CD24. As a result, inhibiting the interaction of CD24-Siglec-10 promotes the phagocytosis of CD24^+^ tumor cells by TAMs, and has been shown to increase survival in preclinical mouse models [16] (Figure 2D). Another approach to stimulate the immune system and target cancer stem cells is the exploitation of engineered third-generation CD24-chimeric antigen receptor (CAR)-expressing natural killer cells (CD24-CAR-NK cells) [98]. CD24-CAR-NK cells are activated following specific binding to the SWA11-antigenic recognition site, and elicit cytotoxic activity against CD24^+^ tumor cell lines (Skov-3, OVCAR3) and patient-derived primary ovarian cancer cells [98]. These studies demonstrate that CD24 is a promising target for molecular targeted therapy, including inducing immune-mediated and antibody-dependent cell cytotoxicity (Figure 2). Consequently, CD24’s tumor specificity should be exploited beyond fluorescent contrast agents.

## 6. Conclusions

Targeted fluorescence image-guided surgery provides additional crucial information about tumor localization, tumor margins, and lymph node infiltration. To improve target specificity and decrease unspecific signals, several biomarkers, such as CA125, EpCAM, VEGF, Her2, integrins, and GnRH receptors, have been tested in preclinical and clinical trial of FIGS, as well as for other diagnostic and therapeutic purposes in EOC. Based on this review, we propose that the small cell surface protein CD24 is an ideal biomarker for FIGS. CD24 meets the criteria of a good imaging biomarker, as it is almost uniquely expressed on tumor cells and predominantly physiologically expressed on developing hematological cells, with low expression in healthy tissues. Its moderate spatial and temporal intratumor heterogenous expression in EOC enables its broad applicability among different stages, in cases of recurrence, and in metastases. Finally, CD24 is expressed on almost 70% of solid tumors, which makes CD24-FIGS suitable for a wide variety of malignancies. In a preclinical study of EOC, CD24-NIR contrast agents were demonstrated to enable the enhanced detection of metastases and an increased resection in heterogeneous CD24-expressing metastatic PDX models. As technology improves, making it possible to produce dyes with favorable optical and physical properties and spurring the development from mAb to antibody fragments and nanobodies, CD24-FIGS has the potential to move to clinical use in EOC and various different cancer debulking and whole organ excision surgeries. We anticipate that a bimodal approach using a CD24-targeted FIGS contrast agent with antibody–drug conjugates, immune blockade, or immunostimulatory CARs may link precision surgical therapy with the postsurgical irradiation of unresected tumor deposits, ultimately benefiting cancer patients.

## Figures and Tables

**Figure 1 jpm-10-00255-f001:**
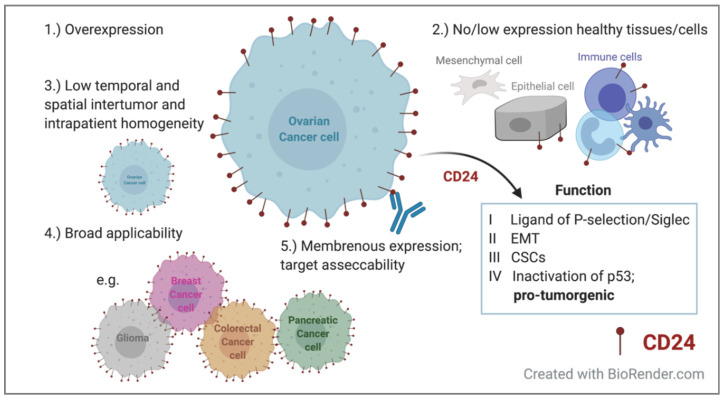
Requirements of a good biomarker for fluorescence image-guided surgery in the context of CD24. It should have (**1**) high tumor specificity with high abundance on tumor cells, and (**2**) the target should be absent in healthy tissue (**3**), while spatial and temporal intratumor heterogeneity should be low [13]. (**4**) It should have broad applicability to a wide variety of malignancies and (**5**) preferably extracellular target expression for better accessibility to widen its clinical applicability. EMT: epithelial–mesenchymal transition; CSCs: cancer stem cells

**Figure 2 jpm-10-00255-f002:**
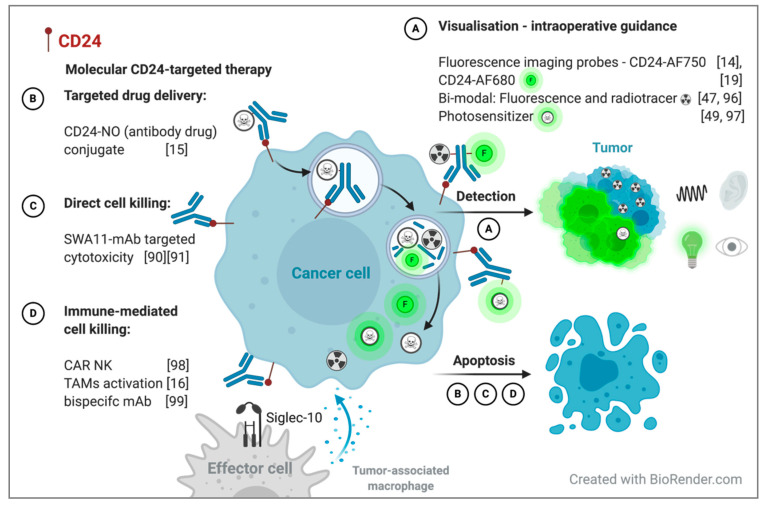
Potential of CD24 as a bimodal imaging biomarker for molecular CD24-targeted therapy (**A**–**C**) and radio- and fluorescent intraoperative guidance (**D**) with dual-labeled CD24 mAbs. A theranostic approach for personalized therapy. NO: nitric oxide, mAb: monoclonal antibody, CAR NK: chimeric antigen receptor-expressing natural killer cells, TAMs: tumor-associated macrophages, AF750: Alexa Fluor 750, AF680: Alexa Fluor 680.

**Table 1 jpm-10-00255-t001:** Molecular biomarkers of ovarian cancer. The present and future.

Biomarker	Preclinical/Clinical	Purpose	Ref.
CA125			
Serum CA125	Clinically approved	Screening	
Oregovomab	Phase I–III	Immunotherapy	[35]
B43.13-IRDye800CW	Preclinical	FIGS	[36]
VEGF			
Bevacizumab	Clinically approved	MTT	*
Bevacizumab-IRDye800CW	Preclinical	FIGS	[37]
EGFR			
Trastuzumab (Her2)	Phase II	MTT	[38]
Trastuzumab-IRDye800CW	Preclinical	FIGS	[37]
Pertuzumab (Her2)	Phase II	MTT	[39,40]
Pertuzumab-IRDye800CW	Preclinical	FIGS	[41]
Her2 nanobody-IRDye800CW	Preclinical	FIGS	[42]
Cetuximab (Her1)	Phase II	MTT	[43]
EPCAM			
Catumaxomab	Phase II/III	Immunotherapy	[44,45]
FRα			
Farletuzumab	Phase III	MTT	[46]
^111^In-farletuzumab-IRDye800CW	Preclinical	FIGS	[47]
Folate-FITC	Phase Ia	FIGS	[10]
EC17	Phase I	FIGS	[11]
OTL38	Phase I–III	FIGS	[12,48]
IR780-loaded folate-targeted nanoparticles	Preclinical	FIGS	[49]

MTT: molecular targeted therapy, FIGS: fluorescence image-guided surgery, CA125: cancer antigen 125, HE4: human epididymis protein 4, FRα: folate receptor alpha, VEGF: vascular endothelial growth factor, EGFR: epidermal growth factor receptor tyrosine kinase family. (* Bevacizumab was approved in 2011 based on the finding of two clinical trials on front-line treatment for high-risk EOC patients (GOG218, ICON7) and in 2012 for recurrent EOC patients based on the results of the GOG213 and OCEANS trials.).

**Table 2 jpm-10-00255-t002:** Expression of CD24 among malignancies.

Malignancy	CD24 Positive	Detection Method	Cohort Size (*n*)	Biomarker	Ref.
Bladder carcinoma					
	75% primary	IHC	60	Therapeutic target	[62]
	(93.3% metastases)				
	63.2%	IHC	125	Predictive	[61]
Breast cancer					
	84.6%	IHC	201	Prognostic	[63]
Colorectal cancer					
	68.7%	IHC	147	Prognostic	[64]
	(84.4% cytoplasmatic)				
	90%	IHC	398	Therapeutic target	[65]
Epithelial ovarian cancer					
	70.1%	IHC	174	Prognostic	[66]
	100% (cytoplasmatic)	IHC	71	Prognostic	[67]
	84%	IHC	56	Prognostic	[68]
	91%	IHC	116	Diagnostic	[18]
	89.7%	IHC	29	Imaging	[19]
Esophageal squamous cell carcinoma					
	40.4%	IHC	151	Prognostic	[69]
Glioma					
	72.8%	IHC (WB, RT-PCR)	151	Prognostic	[70]
Hepatocellular carcinoma					
	66/68%	mRNA	79/31	Diagnostic	[71]
	(91% p53^mut^ HCC)	(Northern Blot, PCR)			
Non-small cell lung carcinoma					
	45%	IHC	89	Prognostic	[72]
Lymphoma					
	91%	TMA	522	/	[73]
Pancreatic cancer					
	26.9%	IHC	67	Prognostic	[74]
	71.6%	IHC	95	Predictive	[75]
Prostate					
	48% primary	IHC	102	Prognostic	[76]
	68% LN metastases		31		
(Clear cell) Renal cell carcinoma					
	41.7%	IHC	108	Prognostic	[77]

IHC: Immunohistochemistry, TMA: tissue microarrays, LN: lymph node, HCC: Hepatocellular carcinoma. Note: IHC staining intensities and the cut-off between positive (medium and high) and negative (weak and absent) are not standardized among the cited studies.

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
