# Peer review of "The Emerging Role of CD24 in Cancer Theranostics—A Novel Target for Fluorescence Image-Guided Surgery in Ovarian Cancer and Beyond"

_jpm, 2020, doi:10.3390/jpm10040255_

Round 1
Reviewer 1 Report
The paper is a well constructed review presenting the applicability of CD24 molecule-conjugated based diagnostic dyes and therapies. The manuscript is clear and well written taking into consideration both the substantive and linguistic level. The only point of my concern is how to obtain an effective level of fluorescence for a time suitable for cytoreductive surgery (4-5 hours or more). This should be shortly discussed by authors.
Author Response
We would like to thank the reviewer for taking the time to evaluate our manuscript and we appreciate the constructive comment, which is a valid and important concern. We agree that it is important to consider an effective level of fluorescence over time for cytoreductive surgery. First, one has to consider the timing of injection of the fluorescence contrast agent prior to cytoreductive surgery. We have discussed that the optimal time point for intraoperative surgery after intravenous injection of the contrast agent should be considered, and it is highly dependent on the size of the conjugate. For small molecules and peptides, the best tumor-to-background ratio is usually achieved within hours, whereas the clearance time for antibody-based contrast agents ranges between two to four days (Tummers et al. 2016, Oncotarget; Kleinmanns et al. 2020, EBioMedicine; Hernot et al. 2019, Lancet Oncol, page 3 line 121-124). In our previous study “CD24-targeted intraoperative fluorescence image-guided surgery leads to improved cytoreduction of ovarian cancer in a preclinical orthotopic surgical model”, we demonstrated the in vivo targeting specificity of CD24-Alexa Fluor 750 in the ovarian tumor xenograft for up to one week after antibody injection (Kleinmanns et al. 2020, EBioMedicine, page 8 line 304-308). The advantage of the long circulation times of antibodies is that it matches well with the half-life of commonly used radiotracers for PET and SPECT, and this bimodal approach in one injected tracer allows for pre-operative imaging as well as intraoperative guidance from both fluorescence and gamma-probe. (Lee et al. 2019 American Journal of Cancer Research, page 10 line 392-397).
The second consideration is to obtain a stable intraoperative fluorescent signal for the duration of surgery. Photobleaching due to excessive excitation should be avoided, as well as instability of the dye, unstable conjugation and fast degradation of the fluorophores. FITC is a fluorophore in the visible light spectrum and it is sensitive to photobleaching, which limits the time available for imaging, but advances in fluorophore development has improved photostability, and photobleaching is less likely to be observed with near infrared fluorophores (Gibbs 2012, Quant Imaging Med Surg.). Gioux et al. 2010 (Mol Imaging) have defined the photobleaching threshold for heptamethine indocyanine class fluorophores and have shown that the FLARE imaging system from Curadel keeps the excitation fluorescence rate below the photobleaching threshold (Page 4 line 139-142).
Reviewer 2 Report
I had the pleasure of reviewing the article entitled: 'The Emerging Role of CD24 in Cancer Theranostics— A Novel Target for Fluorescence Image-guided Surgery in Ovarian Cancer and Beyond'.
The article is very interesting, well written, and easy to read. It is exhaustive in each of its sections and deals with a very interesting topic.
Author Response
We thank the reviewer for the positive evaluation of our manuscript.